# SWI/SNF catalytic subunits' switch drives resistance to EZH2 inhibitors in *ARID1A*-mutated cells

Shuai Wu[1], Nail Fatkhutdinov[1,2], Takeshi Fukumoto[1], Benjamin G. Bitler[1], Pyoung Hwa Park[1], Andrew V. Kossenkov[3], Marco Trizzino[1], Hsin-Yao Tang[4], Lin Zhang[5], Alessandro Gardini[1], David W. Speicher[3,6] & Rugang Zhang [1]

Inactivation of the subunits of SWI/SNF complex such as ARID1A is synthetically lethal with inhibition of EZH2 activity. However, mechanisms of de novo resistance to EZH2 inhibitors in cancers with inactivating SWI/SNF mutations are unknown. Here we show that the switch of the SWI/SNF catalytic subunits from SMARCA4 to SMARCA2 drives resistance to EZH2 inhibitors in *ARID1A*-mutated cells. SMARCA4 loss upregulates anti-apoptotic genes in the EZH2 inhibitor-resistant cells. EZH2 inhibitor-resistant *ARID1A*-mutated cells are hypersensitive to BCL2 inhibitors such as ABT263. ABT263 is sufficient to overcome resistance to an EZH2 inhibitor. In addition, ABT263 synergizes with an EZH2 inhibitor in vivo in ARID1A-inactivated ovarian tumor mouse models. Together, these data establish that the switch of the SWI/SNF catalytic subunits from SMARCA4 to SMARCA2 underlies the acquired resistance to EZH2 inhibitors. They suggest BCL2 inhibition alone or in combination with EZH2 inhibition represents urgently needed therapeutic strategy for *ARID1A*-mutated cancers.

[1] Gene Expression and Regulation Program, The Wistar Institute, Philadelphia, PA 19104, USA. [2] Kazan Federal University, Kazan 420008, Russia. [3] Center for Systems and Computational Biology, The Wistar Institute, Philadelphia, PA 19104, USA. [4] Proteomics and Metabolomics Facility, The Wistar Institute, Philadelphia, PA 19104, USA. [5] Department of Obstetrics and Gynecology, University of Pennsylvania Perelman School of Medicine, Philadelphia, PA 19104, USA. [6] Molecular and Cellular Oncogenesis Program, The Wistar Institute, Philadelphia, PA 19104, USA. These authors contributed equally: Shuai Wu, Nail Fatkhutdinov  Correspondence and requests for materials should be addressed to R.Z. (email: rzhang@wistar.org)

A major discovery of The Cancer Genome Atlas (TCGA) analysis is the identification of genetic alterations in chromatin-modifying factors. The SWI/SNF chromatin remodeling complex is altered in ~20% of human cancers[1], and *ARID1A*, a subunit of this complex, is mutated in up to 62% of ovarian clear cell carcinomas (OCCCs)[2–4]. The SWI/SNF complex remodels nucleosomes to modulate transcription[5]. EZH2 is the catalytic subunit of the polycomb repressive complex 2 (PRC2), which silences its target genes by generating the lysine 27 trimethylation epigenetic mark on histone H3 (H3K27Me3)[6]. Inactivation of the SWI/SNF complex is synthetically lethal with inhibition of EZH2 activity[7]. This is due to antagonistic roles played by the SWI/SNF and PRC2 complex in gene transcription. Highly specific small-molecule EZH2 inhibitors have been developed[8–10]. EZH2 inhibitors are in clinical trials for specific tumor types with known SWI/SNF mutations[7,11]. However, mechanisms of de novo or acquired resistance to EZH2 inhibitors in SWI/SNF-mutated cancer are unknown.

Despite their selectivity and limited toxicity, acquired resistance is a major challenge associated with targeted cancer therapies, including those based on synthetic lethality[12,13]. SMARCA4 (also known as BRG1) and SMARCA2 (also known as BRM) are the catalytic subunits of the SWI/SNF complex[5]. SMARCA4 and SMARCA2 are mutually exclusive in the SWI/SNF complex[5]. Although *SMARCA4*-mutant cells rely on SMARCA2 for proliferation[14,15], there is evidence to suggest the non-redundant roles of SMARCA4 and SMARCA2 in regulating SWI/SNF target genes[16,17].

Here we show that the switch of the SWI/SNF catalytic subunits from SMARCA4 to SMARCA2 drives resistance to EZH2 inhibitors in *ARID1A*-mutated ovarian cancer cells. SMARCA4 decrease dominates over SMARCA2 increase in the switch. SMARCA4 loss leads to suppression of apoptotic pathways through upregulating anti-apoptotic genes such as *BCL2* in the EZH2 inhibitor-resistant (EIR) cells. EIR *ARID1A*-mutated cells are hypersensitive to BCL2 inhibitors such as ABT263. ABT263 is sufficient to overcome resistance to an EZH2 inhibitor and synergizes with an EZH2 inhibitor in vivo in ARID1A-inactivated ovarian tumor mouse models. Together, these data establish that the switch of the SWI/SNF catalytic subunits from SMARCA4 to SMARCA2 underlies the acquired resistance to EZH2 inhibitors, and BCL2 inhibition alone or in combination with EZH2 inhibition represents a novel strategy to overcome and/or prevent EZH2 inhibitor resistance in *ARID1A*-mutated cancers.

## Results

**Catalytic subunits switch in EIR cells**. *ARID1A* mutation typically causes the loss of ARID1A protein expression in OCCCs[2,3,18], and *ARID1A*-mutated OCCC cells such as TOV21G are hypersensitive to EZH2 inhibitors such as GSK126[18–20]. EIR clones were developed by a continuous stepwise exposure to increasing concentrations of GSK126 (Fig. 1a, b), which correlates with resistance to GSK126-induced apoptosis (Supplementary Fig. 1a-b). EIR cells displayed resistance to two additional EZH2 inhibitors that are in clinical trials, while there was no change in sensitivity to chemotherapeutics such as cisplatin or paclitaxel (Supplementary Table 1). RNA-sequencing (RNA-seq) analysis revealed that there were no secondary mutations in *ARID1A* in EIR cells. Consistently, ARID1A and EZH2 expression was not changed in EIR cells (Fig. 1c). The observed resistance was not due to the inability of the EZH2 inhibitor to suppress EZH2 enzymatic activity because H3K27Me3, the enzymatic product of EZH2[6], remained ablated in EIR cells (Fig. 1c). There is evidence to suggest that a decrease in stabilization of the PRC2 complex contributes to intrinsic resistance to EZH2 inhibitors in SWI/

SNF-mutated cells[19]. However, the interaction between EZH2 and SUZ12 was not decreased in the EIR cells (Supplementary Fig. 1c), suggesting that the observed resistance was not due to a decrease in PRC2 stability.

To systematically identify composition changes associated with the SWI/SNF complex in cells with acquired resistance to EZH2 inhibitor, we purified the SWI/SNF complex from parental and EIR cells by immunoprecipitating the core subunit SMARCC1 (also known as BAF155) (Fig. 1d). Liquid chromatography-tandem mass spectrometry (LC-MS/MS) of the pull-down and stoichiometry analysis revealed that there was a significant switch in the catalytic subunit from SMARCA4 to SMARCA2 in EIR cells (Fig. 1e). This switch was validated by co-immunoprecipitation (co-IP) analysis using antibodies to core subunits such as SMARCC1 or SMARCB1 (also known as SNF5) (Fig. 1f, g). In addition, based on sucrose gradient analysis, compared with parental cells, the incorporation of SMARCA4 into the SWI/SNF complex was decreased in EIR cells (Fig. 1h). This was accompanied by an increase in the incorporation of SMARCA2 into the SWI/SNF complex in EIR cells (Fig. 1i). Consistently, RNA-seq analysis revealed that there is a significant upregulation of *SMARCA2* and downregulation of *SMARCA4* in EIR cells. This was validated at both the mRNA and protein levels in these cells (Fig. 1j, k). Together, we conclude that the switch of the catalytic subunits from SMARCA4 to SMARCA2 accompanies the acquired resistance to EZH2 inhibitors in *ARID1A*-mutated cells (Fig. 1l).

**Downregulation of SMARCA4 drives the switch and resistance**. To determine whether SMARCA4 decrease and/or SMARCA2 increase play a role in acquired resistance to EZH2 inhibitors, we knocked down SMARCA4 or ectopically expressed SMARCA2 in TOV21G cells and determined the response to EZH2 inhibitor GSK126. SMARCA4 knockdown, but not SMARCA2 ectopic expression, conferred a decrease in sensitivity to GSK126 (Fig. 2a, b and Supplementary Fig. 2a-b). The decrease in sensitivity to GSK126 induced by SMARCA4 knockdown was observed in multiple *ARID1A*-mutated OCCC cell lines (Supplementary Fig. 2c-h). Conversely, ectopic SMARCA4 expression re-sensitized EIR cells to GSK126 (Fig. 2c, d). This correlated with a concurrent increase in SMARCA2 and a decrease of SMARCA4 in the SWI/SNF complex as determined by anti-BAF155 co-IP analysis (Fig. 2e). Consistently, incorporation of SMARCA4 into the SWI/SNF complex was decreased, while there was an increase in SMARCA2's incorporation into the complex (Fig. 2f). Notably, SMARCA2 knockdown did not affect sensitivity to GSK126 in EIR cells (Supplementary Fig. 2i-j). Consistent with previous reports that SMARCA4 and SMARCA2 concurrent loss is synthetically lethal[14,15], EIR cells were hypersensitive to SMARCA2 knockdown (Supplementary Fig. 2k–m). Together, these data support that SMARCA4 decrease dominates over SMARCA2 increase in the acquired resistance to EZH2 inhibitors.

**SMARCA4 loss promotes an anti-apoptosis gene signature**. Since SMARCA4 loss contributes to the acquired resistance, we next sought to determine the genes directly regulated by SMARCA4 in parental and EIR cells using chromatin immunoprecipitation-sequencing (ChIP-seq) analysis. Toward this goal, we tagged the endogenous SMARCA4 locus with a FLAG epitope using CRISPR because the anti-SMARCA4 antibodies we tested were not robust in ChIP-seq analysis. Consistent with loss of SMARCA4 in EIR cells, there was an overall decrease in SMARCA4 binding in EIR cells compared with parental controls (Fig. 3a). ChIP-seq analysis revealed that *SMARCA4* gene locus is a direct target of SMARCA4 (Fig. 3b), which was

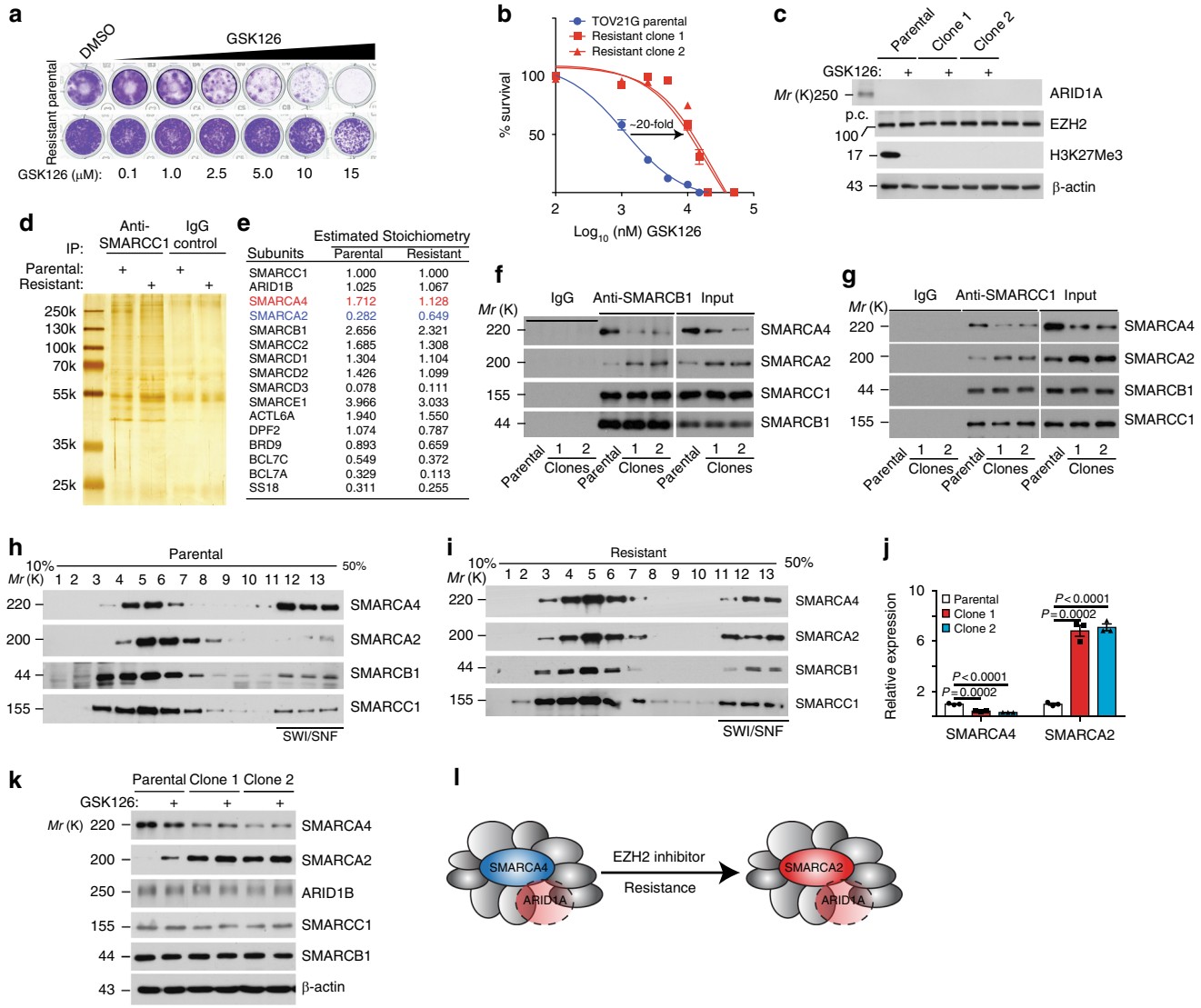

**Fig. 1** The SWI/SNF catalytic subunits' switch from SMARCA4 to SMARCA2 accompanies the de novo resistance to EZH2 inhibitors. **a**, **b** Parental and GSK126-resistant TOV21G cells were subjected to colony formation (**a**) to generate dose response curves to GSK126 (**b**). Arrow points to an ~20-fold increase in GSK126 IC$_{50}$ in the resistant clones. **c** Expression of ARID1A, EZH2, H3K27Me3, and a load control β-actin in the indicated cells passaged with or without 5 μM GSK126 for 3 days determined by immunoblot. p.c. positive control ARID1A wild-type RMG1 cells. **d**, **e** Immunoprecipitation of core SWI/SNF subunit SMARCC1 was separated on a silver stained gel (**d**), or subjected to LC-MS/MS analysis **e**. Stoichiometry of the SWI/SNF subunits identified was normalized to SMARCC1. **f**, **g** Co-immunoprecipitation analysis using antibodies to core subunit SMARCC1 (**f**) or SMARCB1 (**g**) show the switch from SMARCA4 to SMARCA2 in resistant cells. An isotype-matched IgG was used as a control. **h**, **i** Sucrose sedimentation (10–50%) assay of SWI/SNF complex from parental (**h**) or resistant cells (**i**). **j**, **k** Expression of SMARCA4 and SMARCA2 in the indicated cells determined by qRT-PCR (**j**) or immunoblot (**k**). **l** A schematic model: the catalytic subunits from SMARCA4 to SMARCA2 accompanies the de novo resistance to EZH2 inhibitors. Data represent mean ± S.E.M. of three independent experiments (**a**–**c**, **f**–**k**). P-value was calculated via two-tailed t-test

validated by ChIP analysis (Fig. 3c). Therefore, a negative feedback loop contributes to SMARCA4 downregulation in EIR cells (Supplementary Fig. 3a). Consistent with previous reports[20], we showed that SMARCA2 is a target of EZH2/H3K27Me3 (Supplementary Fig. 3b–d), which correlates with the upregulation of SMARCA2 in EIR cells (Fig. 1d, e).

To identify genes that are directly regulated by SMARCA4, we cross-referenced ChIP-seq data with the RNA-seq data comparing gene expression in parental and two individual EIR clones (Fig. 3d). Since we are interested in exploring targets for overcoming resistance to EZH2 inhibitors, we focused on the genes that are upregulated in EIR cells (Fig. 3d). There was a significant enrichment of SMARCA4 target genes in the genes differentially upregulated in EIR cells (Fig. 3d). The analysis

identified a list of 394 direct SMARCA4 target genes upregulated in EIR cells (Fig. 3d). Pathway analysis revealed that the top pathways that are suppressed in EIR cells were cell death and apoptosis (Fig. 3e). Among the cell death/apoptosis signature, the anti-apoptosis gene *BCL2* is a direct SMARCA4 target whose SMARCA4 occupancy in the promoter region was reduced and its expression was significantly upregulated in EIR cells (Fig. 3f and Supplementary Fig. 3e). We validated the upregulation of BCL2 at both the mRNA and protein levels in EIR cells (Fig. 3g, h). In addition, the similar downregulation of SMARCA4 and the accompanying upregulation of SMARCA2 is observed in the *ARID1A*-mutated SKOV3 cells that acquired resistance to GSK126, which correlates with an upregulation of BCL2 in the resistant cells (Supplementary Fig. 3f, g). Finally, we validated the

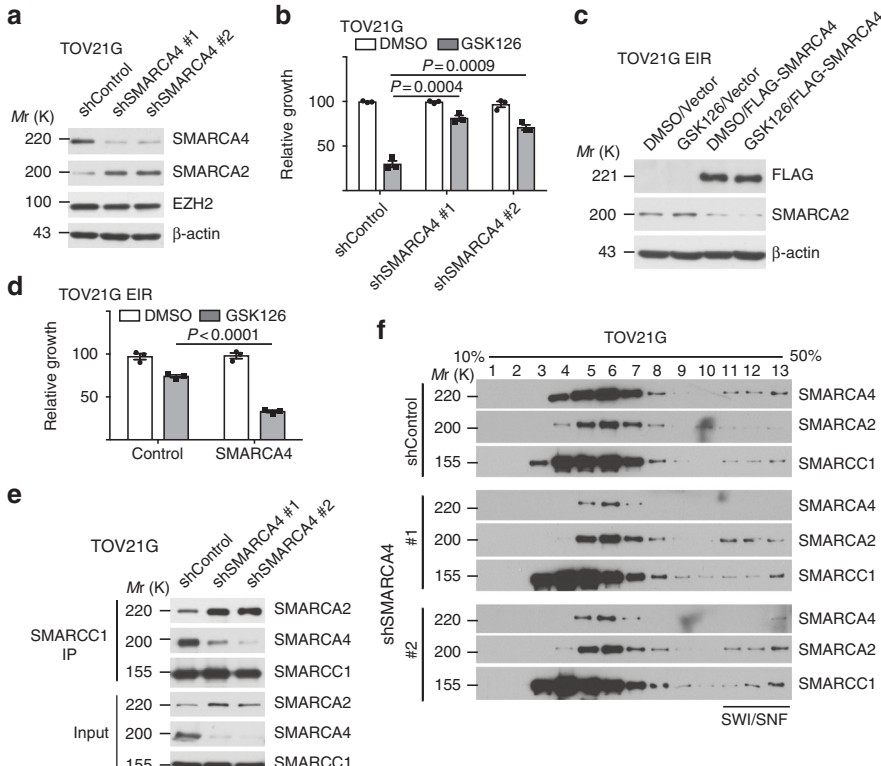

**Fig. 2** Downregulation of SMARCA4 drives the observed switch to SMARC2 in the SWI/SNF complex. **a**, **b** SMARC4 knockdown in parental *ARID1A*-mutated TOV21G cells increases SMARCA2 levels (**a**) and desensitizes parental cells to GSK126 treatment (5 μM) (**b**). **c**, **d** Ectopic SMARCA4 expression in resistant cells decreases SMARCA2 levels (**c**) and resensitizes resistant cells to GSK126 (10 μM) (**d**). **e** Co-immunoprecipitation analysis using an antibody to core subunit SMARCC1 shows the switch of the catalytic subunit from SMARCA4 to SMARCA2 in TOV21G cells with or without SMARCA4 knockdown. **f** Sucrose sedimentation (10–50%) assay of SWI/SNF complex from TOV21G cells with or without SMARCA4 knockdown. Data represent mean ± S.E.M. of three independent experiments (**a**–**e**). *P*-value was calculated via two-tailed *t*-test

decreased association of SMARCA4 with the *BCL2* promoter in EIR cells (Fig. 3i, j). Thus, we conclude that SMARCA4 loss is associated with a decrease in cell death/apoptosis signature in EIR cells.

**ABT263 overcomes the resistance to EZH2 inhibitor in vivo.** We next determined the role of BCL2 in the observed resistance to EZH2 inhibitors. BCL2 knockdown revealed that EIR cells were hypersensitive to BCL2 inhibition (Supplementary Fig. 4a–b). Consistently, EIR cells are sensitive to small-molecule BCL2 inhibitors such as ABT263 and ABT199[21,22] (Supplementary Fig. 4c–d). Both ABT263 and ABT199 are BCL2 inhibitors that are in clinical trials, while ATB263 is less selective because it also inhibits other members of BCL2 family members such as BCL-xL[21,23]. Consistent with previous reports[24], ABT199-treated EIR cells upregulated BCL-xL (Supplementary Fig. 4e). In addition, SMARCA4 loss leads to a decrease in cell death/apoptosis signature. These findings indicate that ABT263 is advantageous in this context. Markers of apoptosis such as cleaved caspase 3, cleaved PARP p85, and Annexin V were significantly induced by ABT263 in EIR cells compared with controls (Fig. 4a, b). In addition, SMARCA4 knockdown sensitized the *ARID1A*-mutated cells to ABT263 (Supplementary Fig. 4f). We next determined whether ABT263 and GSK126 synergize in suppressing the growth of *ARID1A*-inactivated cells. Indeed, ABT263 and GSK126 were synergistic in suppressing the growth of multiple *ARID1A*-mutated OCCC cells (Fig. 4c and Supplementary Fig. 4g–j). Notably, the observed synergy is ARID1A-status-dependent because the synergy was observed in *ARID1A* knockout

but not in control wild-type cells (Supplementary Fig. 4g–h). A similar observation was also made in primary cultures of OCCC cells with or without ARID1A expression (Supplementary Fig. 4k–m). Indeed, IC$_{50}$ of ABT263 is significantly lower in ARID1A-deficient compared with ARID1A-proficient OCCC cells (Supplementary Fig. 4n). ARID1A ChIP-seq in *ARID1A* wild-type cells revealed that *BCL2* is a direct ARID1A target (Supplementary Fig. 4o), which was validated by ChIP analysis (Supplementary Fig. 4p). ARID1A knockout upregulated BCL2 expression in *ARID1A* wild-type cells (Supplementary Fig. 4q). We conclude that *BCL2* is a ARID1A target gene (Supplementary Fig. 4r).

Orthotopic tumors formed by EIR cells were resistant to EZH2 inhibitor GSK126 (Fig. 4d and Supplementary Fig. 5a). Significantly, ABT263 was sufficient to cause the regression of the established EIR OCCCs (Fig. 4d, e). This correlated with a significant improvement of survival of mice bearing the EIR OCCCs (Fig. 4f). As a control, GSK126 suppressed the growth of parental TOV21G cells (Supplementary Fig. 5b). The dose of ABT263 used in this study did not significantly affect the body weight of treated mice (Supplementary Fig. 5c), suggesting that effective doses can be achieved with minimal toxicity. Notably, ABT263 and GSK126 were synergistic in suppressing the growth of xenograft tumors formed by ARID1A-deficient primary OCCC cultures (Fig. 4g and Supplementary Fig. 5d). In addition, ABT263 and GSK126 combination significantly suppressed the growth of the established tumors in the immunocompetent conditional genetic *Arid1a*$^{-/-}$/*Pik3ca*$^{H1047R}$ mouse OCCC model (Supplementary Fig. 5e).

We next performed immunohistochemical (IHC) staining for markers of cell proliferation (Ki67), apoptosis (cleaved caspase 3),

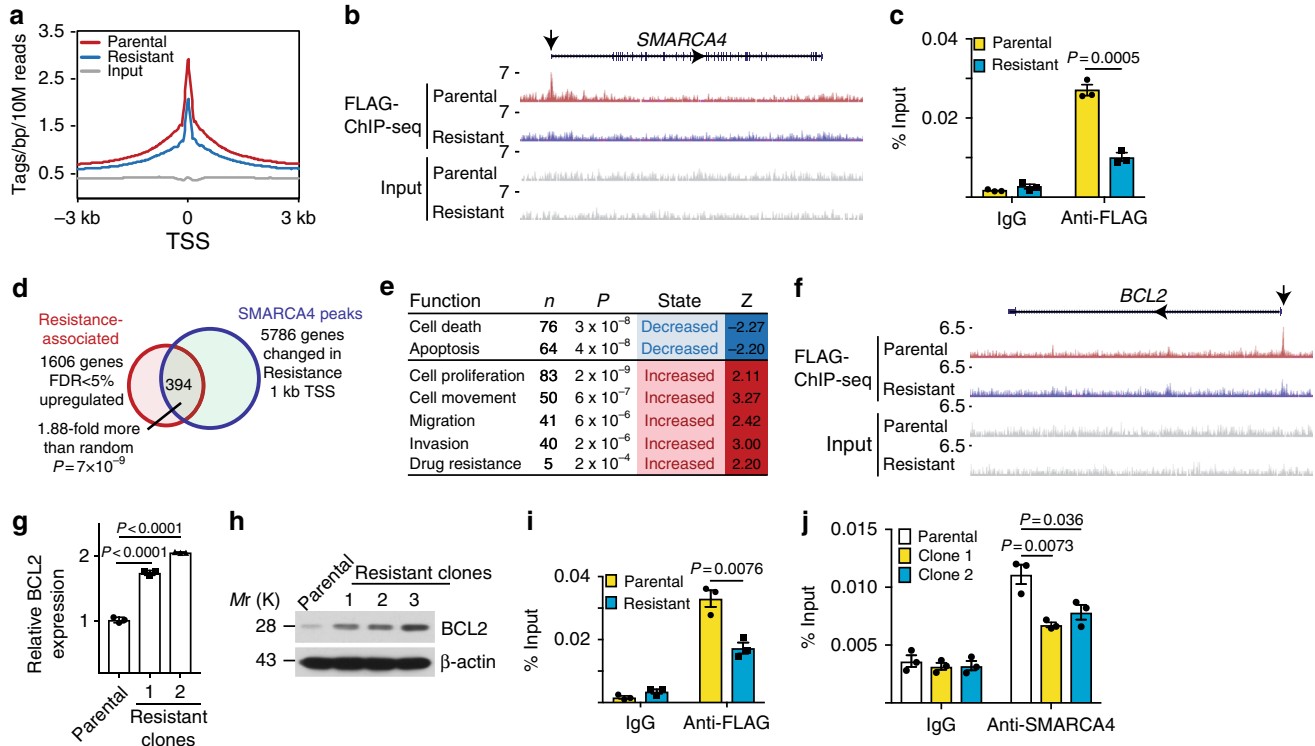

**Fig. 3** SMARCA4 loss promotes resistance to EZH2 inhibitors by upregulating an anti-apoptosis gene signature. **a** ChIP-seq profiles of SMARCA4 in parental and resistant cells. TSS: transcription starting sites. **b** ChIP-seq tracks of SMARCA4 on its own promoter region in endogenously FLAG-tagged parental and resistant cells. Arrow points to the loss of SMARCA4 binding in its own promoter region. **c** ChIP-qPCR validation of a decrease of SMARCA4 binding to its own promoter. **d** Venn diagram showing the genome-wide overlap analysis between SMARCA4 ChIP-seq and genes upregulated in RNA-seq in parental and resistant cells. **e** Top pathways enriched among the genes identified in **d**. **f** ChIP-seq tracks of SMARCA4 on the *BCL2* promoter region in endogenously FLAG-tagged parental and resistant cells. **g**, **h** qRT-PCR (**g**) and immunoblot (**h**) of BCL2 levels in parental and resistant cells. **i**, **j** ChIP-qPCR validation of a decrease in SMARCA4 binding on the *BCL2* promoter in resistant cells using antibodies against endogenously tagged FLAG (**i**) or endogenous SMARCA4 (**j**). Data represent mean ± S.E.M. of three independent experiments (**c**, **g**–**j**). *P*-value was calculated via two-tailed *t*-test

EZH2, and H3K27Me3 in the dissected EIR tumors treated with GSK126 or ABT263 (Fig. 4h). Consistent with in vitro mechanistic findings, ABT263 significantly suppressed Ki67 and increased cleaved caspase 3, without changing the expression of EZH2 or H3K27Me3 (Fig. 4h, i). In contrast, GSK126 significantly decreased H3K27Me3 in these tumors without affecting the expression of EZH2, Ki67, or cleaved caspase 3 (Fig. 4h, i). As controls, both GSK126 and ABT263 decreased Ki67 and cleaved caspase 3 in tumor formed by parental TOV21G cells (Supplementary Fig. 5f-g). Thus, we conclude that ABT263 regressed EIR OCCC and improved the survival of the tumor-bearing mice through suppressing proliferation and promoting apoptosis in vivo. They also suggest that ABT263 and GSK126 combination may prevent the development of resistance to EZH2 inhibitors.

## Discussion

Here we show that the SWI/SNF catalytic subunits switch from SMARCA4 to SMARCA2 underlies the de novo resistance to EZH2 inhibitors. Notably, we discover that SMARCA4 is self-regulated due to the binding of SMARCA4 to its own gene promoter. Therefore, *SMARCA4* downregulation occurs at the transcriptional level through a negative feedback loop. It has previously been shown that there is a reciprocal assembly of SMARC2 into SWI/SNF complexes when SMARCA4 is downregulated[14]. We show that SMARCA2 is upregulated while SMRC4 is downregulated in the EIR cells. Thus, our data suggest that the exchange from SMARCA4 to SMARC2 in SWI/SNF

complexes occurs due to the reciprocal regulation and assembly between these two catalytic subunits.

We show that ABT263 alone at a dose with minimal toxicity was sufficient to cause the regression of the established EIR tumors. Thus, this represents a novel strategy to overcome and/or prevent the development of de novo resistance to EZH2 inhibitors in *ARID1A*-mutated cancers. Indeed, ABT263 and GSK126 are synergistic in suppressing the growth of ARID1A-inactivated primary OCCC in vivo in a xenograft model. In *ARID1A*-mutated cells, SMARCA4 loss drives acquired resistance to EZH2 inhibition, which correlates with an increase in SMARCA2 due to EZH2 inhibition. Interestingly, *SMARCA4*-mutated cells are hypersensitive to EZH2 inhibition[20,25,26]. Thus, the genetic context in which mutations in the SWI/SNF complex occur should be taken into consideration in EZH2 inhibitor trails. Indeed, ~42% of *ARID1A*-mutated OCCCs also simultaneously harbor mutations in *SMARCA4*[4]. Our data suggest that these patients may not respond to EZH2 inhibitor as a single agent but will likely respond to a combination of EZH2 and BCL2 inhibition.

In summary, we report the first de novo EIR mechanism in the context of SWI/SNF subunit *ARID1A* mutation. We discovered a potential therapeutic strategy for overcoming acquired resistance to EZH2 inhibitors. In addition, our data demonstrated that BCL2 inhibitors alone or in combination with EZH2 inhibitors may represent therapeutic strategies for *ARID1A*-mutated cancers. Given that the SWI/SNF subunits are among the most frequently mutated genes in human cancers[1,27] and EZH2 inhibitors are in clinical trials for tumors with mutations in the SWI/SNF

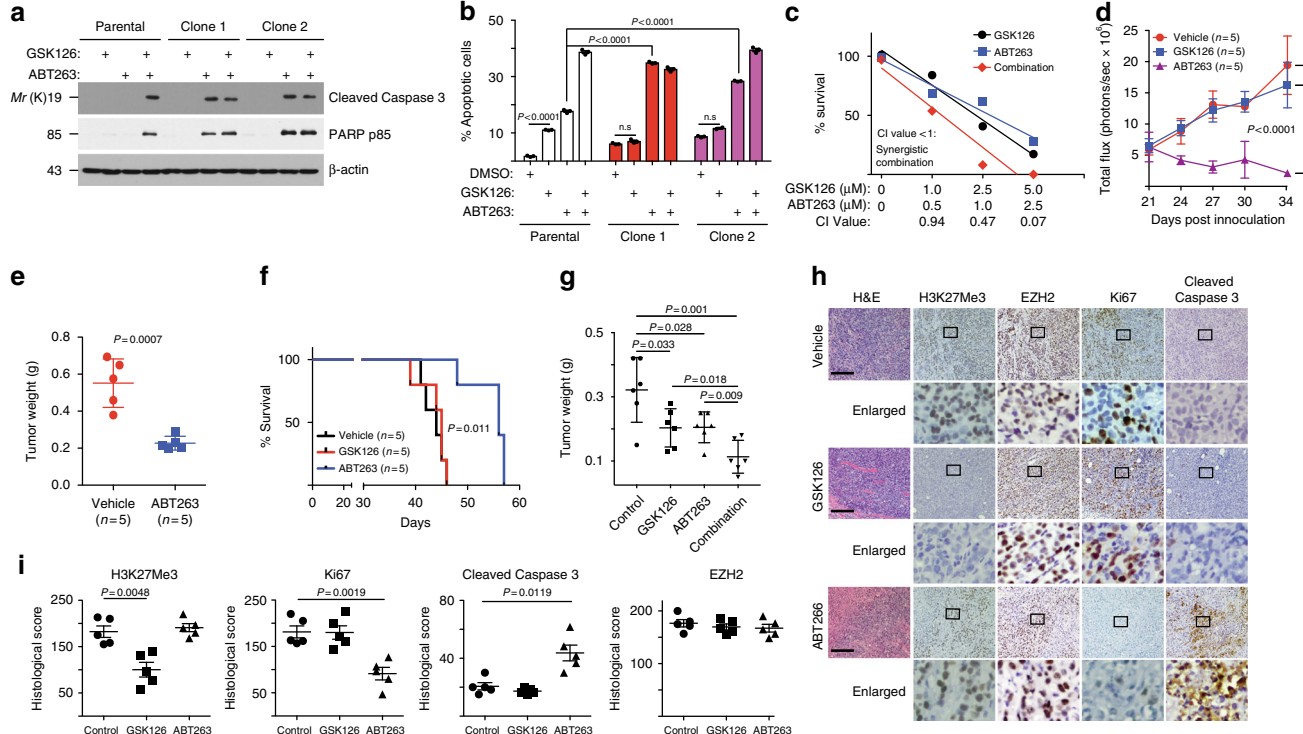

**Fig. 4** ABT263 overcomes de novo resistance to the EZH2 inhibitor. **a**, **b** Parental and resistant TOV21G cells were treated with 0.5 μM ABT263, 5 μM GSK126, or in combination. Expression of markers of apoptosis were analyzed by immunoblot (**a**) or quantified by Annexin V staining (**b**). **c** Synergy analysis between GSK126 and ABT263 in TOV21G cells. **d**–**f** ABT263 regresses the established EZH2 inhibitor-resistant TOV21G orthotopically transplanted tumors (**d**, **e**) and improves the survival of the tumor-bearing mice (**f**) in vivo. **g** GSK126 and ABT263 are synergistic in suppressing the growth of the xenograft ovarian tumors formed by primary OCCC cultures. **h**, **i** Immunohistochemical staining for EZH2, H3K27me3, Ki67, and cleaved caspase 3 using consecutive sections of the dissected tumors from the indicated treatment groups (**h**). Bar = 100 μM. Histological score (H score) calculated for the indicated staining (**i**). Data represent mean ± S.E.M. of three independent experiments (**a**–**c**). P-value was calculated via two-tailed t-test

complex, these findings have important implications for developing therapies for these tumors.

## Methods

**Cell culture conditions and transfection**. The protocol for using primary cultures of human ovarian clear cell tumor cells was approved by the University of British Columbia Institutional Review Board. Informed consent was obtained from human subjects. All relevant ethical regulations have been complied with. The primary tumor cells were cultured in RPMI 1640 supplemented with 10% fetal bovine serum (FBS) and 1% penicillin/streptomycin[28]. Ovarian clear cell carcinoma cell lines TOV21G, OVISE, OVCA429, OVTOKO, and SKOV3 were cultured in RPMI 1640 with 10% FBS and 1% penicillin/streptomycin at 37 °C supplied with 5% CO$_2$. Ovarian clear cell carcinoma cell line RMG1 cells were cultured in 1:1 Dulbecco's modified Eagle's medium (DMEM)/F12 supplemented with 10% FBS. Ovarian clear cell carcinoma cell line ES2 cells were cultured in DMEM with 10% FBS and 1% penicillin/streptomycin at 37 °C supplied with 5% CO$_2$. Primary ovarian cancer cells were cultured in RPMI 1640 with 10% FBS and 1% penicillin/streptomycin at 37 °C supplied with 5% CO$_2$. Viral packing cells 293FT and Phoenix were cultured in DMEM with 10% FBS and 1% penicillin/streptomycin at 37 °C supplied with 5% CO$_2$. All the cell lines were authenticated at The Wistar Institute Genomics Facility using short tandem repeat DNA profiling. Mycoplasma testing was performed using LookOut Mycoplasma PCR detection (Sigma) every month. Three-dimensional culture followed previous methods[18]. Transfection was performed using Lipofectamine 2000 (Life Technologies) following the manufacturer's specifications. Each of the experiments was performed in triplicate in three independent experimental repeats unless otherwise stated.

**Reagents**. GSK126 was purchased from XcessBio for in vitro experiments and from Active Biochem for in vivo experiments. EPZ-6438, CPI-169, Paclitaxel, and Cisplatin were purchased from Selleckchem. ABT263 and ABT199 were purchased from ApexBio. Growth factor-reduced Matrigel was purchased from Corning. pLKO.1-sh*BRG1* (TRCN0000015549 and TRCN0000015552), pLKO.1-sh*BRM* (TRCN0000020329 and TRCN0000020332), and pLKO.1-sh*BCL2* (TRCN0000040069 and TRCN0000040071) were obtained from Open Biosystems. *SMARCA4* plasmid (#1959) was obtained from Addgene and *SMARCA4* gene was

cloned into pLVX system (Clontech). *SMARCA2* lentivirus plasmid was ordered from Genecopoeia (EX-M0272-Lv03).

The following antibodies were obtained from the indicated suppliers: rabbit anti-ARID1A (Abcam, cat. no. ab182560, 10 μg/IP for ChIP or ChIP-seq; Cell Signaling, cat. no. 12354, 1:1000 for western blot); mouse anti-ARID1B (Abgent, cat. no. AT1189a, 1:1000 for western blot); rabbit anti-EZH2 (Cell Signaling, cat. no. 5246, 2 μg/IP for IP, 5 μg/IP for ChIP, 1:2000 for IHC, and 1:2000 for western blot); rabbit anti-H3K27me3 (Cell Signaling, cat. no. 9733, 5 μg/IP for ChIP, 1:2000 for IHC, and 1:1000 for western blot); rabbit anti-SUZ12 (Bethyl, cat. no. A302-407A, 1:1000 for western blot); mouse anti-BRG1 (Santa Cruz, cat. no. sc-17796, 10 μg/IP for ChIP); rabbit anti-BRG1 (Cell Signaling, cat. no. 49360, 1:1000 for western blot); rabbit anti-BRM (Cell Signaling, cat. no. 11966, 5 μg/IP for ChIP and 1:1000 for western blot); goat anti-BAF155 (Santa Cruz, cat. no. sc-9746, 2 μg/IP for IP); rabbit anti-BAF155 (Cell Signaling, cat. no. 11956, 1: 1000 for western blot); rabbit anti-Pol II (Santa Cruz, cat. no. sc-47701, 5 μg/IP for ChIP); rabbit anti-SNF5 (Bethyl, cat. no. A301-087A, 5 μg/IP for IP and 1:1000 for western blot); mouse anti-SNF5 (Abcam, cat. no. ab42503, 1:1000 for western blot); mouse anti-BCL2 (Cell Signaling, cat. no. 15071, 1:1000 for western blot); rabbit anti-cleaved caspase 3 (Cell Signaling, cat. no. 9661, 1:1000 for western blot and 1:200 for IHC); rabbit anti-PARP p85 (Promega, cat. no. G7341, 1:1000 for western blot); mouse anti-Flag M2 (Sigma, cat. no. F1804, 5 μg/IP for ChIP and ChIP-seq, and 1:2000 for western blot); mouse anti-β-actin (Sigma, cat. no. A5316, 1:5000 for western blot); and mouse anti-Ki67 (Cell Signaling, cat. no. 9449, 1:1000 for IHC).

**Lentivirus infection**. pLKO.1-shRNA and pLVX system were used for lentivirus package. HEK293FT cell was transfected by Lipofectamine 2000. Lentivirus was harvested and filtered with 0.45 μm filter 48 h post transfection. Cells infected with lentiviruses were selected in 1 μg/ml puromycin 48 h post infection.

**Western blot and IP**. Protein was extracted with RIPA lysis buffer (50 mM Tris (pH 8.0), 150 mM NaCl, 1% Triton X-100, 0.5% sodium deoxycholate, and 1 mM phenylmethylsulfonyl fluoride (PMSF)). Samples were separated by SDS-polyacrylamide gel electrophoresis (SDS-PAGE) and transferred to polyvinylidene fluoride membrane (Millipore). Membranes were blocked with 5% non-fat milk and then incubated with primary antibodies and secondary antibodies.

Unprocessed images of scanned immunoblots shown in Figures and Supplementary Figures are provided in Supplementary Fig. 6.

For IP, nuclear fractions were prepared by ammonium sulfate precipitation[1]. Briefly, cells were collected and resuspended in buffer A (10 mM HEPES (pH 7.6), 10 mM KCl, 25 mM, 10% glycerol, 1 mM dithiothreitol (DTT), and 1 mM PMSF) on ice. Nuclei were harvested by centrifugation (1300 × g for 4 min) and lysed by 0.3 M ammonium sulfate in buffer C (10 mM HEPES (pH 7.6), 3 mM MgCl₂, 100 mM KCl, 0.1 mM EDTA, 10% glycerol, 1 mM DTT, and 1 mM PMSF). Soluble nuclear proteins were separated by ultracentrifugation (100 000 × g for 30 min) and precipitated with 0.3 g/ml ammonium sulfate for 30 min on ice. Protein precipitate was isolated by ultracentrifugation (100 000 × g for 30 min) and resuspended in IP lysis buffer (50 mM Tris-HCl (pH 8.0), 150 mM NaCl, 1% NonidetP-40, 0.5% deoxycholate, 1 mM DTT, and 1 mM PMSF) for IP.

**Silver staining and mass spectrometry.** Endogenous complexes were purified following the IP protocol. Quality of the samples was determined by western blot and silver staining. Briefly, the SDS-PAGE gel was fixed in fixation buffer 1 (50% methanol and 10% acetic acid) for at least 15 min and fixation buffer 2 (10% methanol and 7% acetic acid) for 1–2 h. The gel was washed with 1:10 gluteraldehyde (25%):water solution for 15 min and then three times with deionized water for 15 min. The staining solution was prepared by dropping solution B (1 g AgNO₃ in 5 ml deionized water) into solution A (0.185 ml 10 M NaOH, 2.8 ml NH₄OH, and 22.5 ml deionized water). The final volume was brought to 100 ml by adding 70 ml deionized water. The gel was stained for 15 min and washed three times with deionized water for 2 min. The stain was developed with developing solution (0.5 ml 1% citric acid and 0.05 ml 38% formaldehyde in 100 ml deionized water) to appropriate signal and then stopped by stop solution (50% methanol and 5% acetic acid) for 10 min.

The samples were digested with trypsin and analyzed by LC-MS/MS on a Q Exactive Plus mass spectrometer. MS/MS spectra generated from the LC-MS/MS runs were searched using full tryptic specificity against the UniProt human database using the MaxQuant 1.5.2.8 program. Protein quantification was performed using unique + razor peptides. Razor peptides are shared (non-unique) peptides assigned to the protein group with the most other peptides (Occam's razor principle). False discovery rates (FDRs) for protein and peptide identifications were set at 1%. The stoichiometry of the interaction was normalized against SMARCC1 intensity value.

**Sucrose density sedimentation assay.** Nuclear fractions were prepared following the IP protocol, except the nuclear protein precipitate was resuspended in HEMG-0 buffer (25 mM HEPES (pH 7.9), 0.1 M EDTA, 12.5 mM MgCl₂, and 100 mM KCl). Sucrose density sedimentation assay was modified from previous methods[29]. Briefly, 1 mg of nuclear protein was carefully overlaid onto a 5-ml 20–50% sucrose gradient (in HEMG-0 buffer) prepared in a 5-ml 13 × 51 mm polyallomer centrifuge tube (Beckman Coulter, 326819). Tubes were placed in a SW-55 Ti swingbucket rotor and centrifuged at 4 °C for 16 h at 100 000 × g. Fractions (0.4 ml) were collected for immunoblotting analyses.

**Reverse-transcriptase quantitative PCR.** Total RNA was isolated using Trizol (Invitrogen) according to the manufacturer's instruction. Extracted RNAs were used for reverse-transcriptase PCR (RT-PCR) with High-Capacity cDNA Reverse Transcription Kit (Thermo fisher). Quantitative PCR (qPCR) was performed using QuantStudio 3 Real-Time PCR System. The primers sequences used for quantitative RT-PCR are as follows: SMARCA4 forward: 5′-CTTATGGT-CAATGGTGTC-3′ and reverse: 5′-GTTCAGGTTGTTGTTGTA-3′; SMARCA2 forward: 5′-AGTATGTAGCCAATCTGA-3′ and reverse: 5′-CTCCTCTTCTTCTTCTCT-3′; beta-microglobulin (B2M) forward: 5′-GGCATTCCTGAAGCTGACA-3′ and reverse: 5′-CTTCAATGTCGGATGGAT-GAAAC-3′; BCL2 forward: 5′-TGCCTTTGTGGAACTGTA-3′ and reverse: 5′-GAGCAGAGTCTTCAGAGA-3′. B2M was used as an internal control.

**Colony formation.** Two-dimensional colony formation was adapted from published methods[28]. Briefly, cells were seeded in 24-well plates with different number according to the growth rate. Cell medium was changed every 3 days with appropriate drug doses for 12 days. Colonies were visualized by staining the plates with 0.05% crystal violet. Integrated density was determined using NIH ImageJ software. Colonies were also dissolved by 10% methanol and 10% acetic acid and quantified by absorbance at 570 nm using BioTek microplate reader.

**Annexin V/propidium iodide staining.** Apoptosis was detected using an Annexin V fluorescein isothiocyanate and propidium iodide (PI) kit (Thermo Fisher, V13242) following the manufacturer's instructions. Briefly, cells were washed with cold phosphate-buffered saline (PBS) and resuspended in Annexin V binding buffer and stained with Annexin V and PI at room temperature for 15 min and then analyzed immediately. Results were analyzed with FlowJo version 7 software module.

**Construction of endogenously Flag-tagged SMARCA4 cell line.** PX458 (Addgene #48138) and pUC19 containing Flag-P2A-puro were used to construct endogenously Flag-tagged SMARCA4 cell line. Briefly, gRNA (GGGTCGAGACT GGAATGTCG) targeting the 3′ end of SMARCA4 coding region was inserted into PX458. About 500 bp homologous arms at both sides of gRNA targeting site were cloned and inserted into both sides of Flag-P2A-puromycin (XhoI/EcoRV and XbaI/PstI). The primers used for cloning the homolog recombination arms are following: SMARCA4 left arm forward: 5′-TCCTCGAGCCAAGATGGTTTGGA AGCTGTAGGTC-3′ (XhoI) and reverse: 5′-CTGATATCGTCTTCTTCGCTGC CACTTCCTGAGCGG-3′ (EcoRV); SMARCA4 right arm forward: 5′- CCTCT AGACATTCCAGTCTCGACCCCGAGCCCCT-3′ (XbaI) and reverse: 5′-GCC TGCAGGTGCGTTTTGTTGTTGGTTTAATTTATTACTG-3′ (PstI). Cells were co-transfected with two plasmids and then selected with 1 μg/ml puromycin. Single colonies were picked up and expanded for ChIP-seq or ChIP-qPCR.

**Chromatin immunoprecipitation.** Cells were crosslinked with 1% formaldehyde for 10 min at room temperature. The reaction was quenched by 0.125 M glycine for 5 min. For SMARCA2, SMARCA4 ChIP, or endogenously Flag-tagged SMARCA4 ChIP, dual crosslinking was utilized. Cells were washed with PBS twice and crosslinked with 2 mM ethylene glycol bis(succinimidyl succinate) (EGS, Thermo Fisher, 21565) for 20 min and then crosslinked with 1% formaldehyde for another 10 min, followed by 0.125 M glycine quenching for 5 min.

Fixed cells were lysated with ChIP lysis buffer 1 (50 mM HEPES-KOH (pH 7.5), 140 mM NaCl, 1 mM EDTA (pH 8.0), 1% Triton X-100, and 0.1% DOC) on ice and lysis buffer 2 (10 mM Tris (pH 8.0), 200 mM NaCl, 1 mM EDTA, and 0.5 mM EGTA) at room temperature. Chromatin was digested with MNase in digestion buffer (10 mM Tris 8.0, 1 mM CaCl₂, and 0.2% Triton X-100) at 37 °C for 15 min. The nucleus was broken down by one pulse of bioruptor with high output.

The following antibodies were used for ChIP: mouse anti-BRG1 (Santa Cruz, cat. no. sc-17796); rabbit anti-ARID1A (Abcam, cat. no. ab182560); rabbit anti-EZH2 (Cell Signaling, cat. no. 5246); rabbit anti-H3K27Me3 (Cell Signaling, cat. no. 9733); rabbit anti-BRM (Cell Signaling, cat. no. 11966); mouse anti-Flag M2 (Sigma, cat. no. F1804); and rabbit anti-Pol II (Santa Cruz, cat. no. sc-47701). Chromatin was incubated overnight at 4 °C and protein A + G Dynabeads were added to the reaction for another 1.5 h. Magnetic beads were washed and chromatin was eluted and reversed. Chromatin was then treated with proteinase K and purified with Gel extraction kit (Qiagen, cat. no. 28706). ChIP DNA was used for ChIP-qPCR or ChIP-seq.

For ChIP-qPCR, the following primers were used: SMARCA4 locus forward: 5′-TACAGTCGCCCTCCCAATTA-3′ and reverse: 5′-ATCGCAGCTTCGCCAAA-3′; SMARCA2 locus forward: 5′-ATTGGTAGGCAGGCCTTTAGGCAA-3′ and reverse: 5′-GGTACCAGAGGCAGGGA-3′; BCL2 locus forward: 5′-CCCATCAATCTTCAGCACTCT-3′ and reverse: 5′-GGGAATCGATCTGGAAATCCTC-3′.

**IHC staining.** IHC staining was performed on consecutive sections[28]. Tissue sections were stained using Dako EnVision+ system following the manufacturer's instructions. Briefly, formalin-fixed, paraffin-embedded tumors were sectioned and slides were deparaffinized using xylenes (Fisher Scientific, cat. no. 1330-20-7). Antigens were unmasked using citrate buffer (Thermo Fisher, 005000). Endogenous peroxidases were quenched with 3% hydrogen peroxide in methanol. Staining was performed using antibodies against H3K27Me3 (Cell Signaling, cat. no. 9733, 1:2000), EZH2 (Cell Signaling, cat. no. 5246, 1:2000), cleaved caspase 3 (Cell Signaling, cat. no. 9661, 1:200), or Ki67 (Cell Signaling, cat. no. 9449, 1:1000). Counterstaining was performed using Mayer's Hematoxylin (Dako, cat. no. 3309S). Expression of the stained markers was scored using a histologic score (H score).

**In vivo animal model.** The protocols were approved by the Institutional Animal Care and Use Committee of the Wistar Institute. Results from in vitro experiments were used to determine the in vivo sample size. Intrabursal model was performed as described[28]. Briefly, 1 × 10⁶ luciferase-expressing TOV21G cells and EIR TOV21G cells were unilaterally injected into the ovarian bursa sac of 6- to 8-week-old female NSG mice. Tumors were visualized by injecting luciferin (intraperitoneal, 4 mg/mouse) resuspended in PBS and imaged with an In Vivo Imaging System twice a week. Mice were randomized into three group based on luciferase activity 3 weeks after injection and treated with vehicle control (captisol and 4% dimethylsulfoxide (DMSO)/4% Tween 20/92% PBS), GSK126 (50 mg/kg in captisol and 4% DMSO/4% Tween 20/92% PBS daily), or ABT263 (50 mg/kg in 4% DMSO/4% Tween 20/96% PBS and captisol daily) for 2 weeks. The same experimental procedure was performed for parental TOV21G cells except the mice were randomized 1 week after orthotopic transplantation. Tumor growth was monitored by measuring luciferase twice a week. Images were analyzed using Live Imaging 4.0 software. Imaging analysis was performed blindly but not randomly. At the end of the experiments, tumors were surgically dissected and tumor burden was calculated on the basis of tumor weight.

For primary OCCC subcutaneous model, 1 × 10⁶ ARID1A-deficient XVOA295 primary OCCC cultures were subcutaneously injected into 6- to 8-week-old female NSG mice. Mice were randomized into four groups (n = 6/group) and treated with vehicle control (captisol and 4% DMSO/4% Tween 20/92% PBS), GSK126 (50 mg/

kg in captisol and 4% DMSO/4% Tween 20/92% PBS daily), ABT263 (50 mg/kg in 4% DMSO/4% Tween 20/96% PBS and captisol daily), or combination of GSK126 and ABT263 for 2 weeks. Tumor size was measured twice a week. Following treatment, mice were euthanized and tumors were surgically dissected and tumor burden was calculated on the basis of tumor weight.

For $Arid1a^{-/-}/Pik3ca^{H1047R}$ genetic clear cell ovarian tumor mouse model, the transgenic mice were generated by crossing $Arid1a^{flox/flox}$ mice with $R26$-$Pikca^{H1047R}$ (Jackson Laboratory, Jax no. 016977)[28]. All mice were maintained in specific pathogen-free barrier facilities. Administration of intrabursal adeno-Cre was used to induce OCCC[28]. Mice were randomized into two groups 5 weeks after adeno-Cre injection and treated with vehicle control or a combination of GSK126 (50 mg/kg, daily) and ABT263 (50 mg/kg, daily) for 21 days. Following treatment, mice were euthanized and tumors were surgically dissected and tumor burden was calculated on the basis of tumor weight.

**Sequencing and bioinformatics.** For RNA-seq, extracted RNAs were digested with DNase I (Qiagen, 79254) and purified using RNeasy MinElute Cleanup Kit (Qiagen, 74106). RNA-Seq libraries were prepared using TruSeq Total RNA Sample Prep Kit (Illumina) and sequenced with Illumina NextSeq 500 using 75 bp paired-end run. For ChIP-seq, 10 ng ChIP DNA was used for library construction. NEBNext Ultra DNA Library Prep Kit (NEB, E7645) was used to prepare sequencing library. The libraries were sequenced in a 75 bp single-end run using Illumina NextSeq 500.

RNA-seq data were aligned using bowtie2[30] against hg19 version of the human genome and RSEM v1.2.12 software[31] was used to estimate raw read counts and RPKM using Ensemble gtf tracks. DESeq2[32] was used to estimate significance of differential expression between parental TOV21G vs treated and untreated resistant clone samples. Overall gene expression changes were considered significant if passed FDR < 5% thresholds. CHIP-seq data were aligned using bowtie[33] against hg19 version of the human genome and HOMER[34] was used to call significant peaks in TOV21G sample using resistant clone sample as a control and in resistant clone sample using TOV21G as a control using –style histone option and peaks that passed FDR < 5% threshold were considered as significantly different between TOV21G and resistant clones. Genes that had at least one significant BRG1 peak within 1 kb from transcription starting sites were considered and overlapped with genes significantly upregulated in resistant clones. Significance of overlap was tested using hypergeometric test using 57 736 Ensemble genes as a population size. Gene-set enrichment analysis of gene sets was done using QIAGEN's Ingenuity® Pathway Analysis software (IPA®, QIAGEN, Redwood City, www.qiagen.com/ingenuity) using "Diseases & Functions" analysis. Significantly enriched cell line nonspecific functions with $p < 10^{-6}$ that had a significantly predicted activation state ($|Z| > 2$) were reported. The RNA-seq and ChIP-seq data were submitted to the Gene Expression Omnibus database and can be accessed using accession number: GSE110450.

**Statistical analysis and reproducibility.** Statistical analysis was performed using GraphPad Prism 7 (GraphPad) for Mac OS. Experiments were repeated three times unless otherwise stated. The representative images were shown unless otherwise stated. Quantitative data are expressed as mean ± S.E.M. unless otherwise stated. Analysis of variance with Fisher's least significant difference was used to identify significant differences in multiple comparisons. Combination index was analyzed by Compsyn software. Imaging analysis was performed blindly but not randomly. Animal experiments were randomized. There was no exclusion from the experiments.

## Data availability

All sequencing data have been deposited in the Gene Expression Omnibus (GEO) under accession GSE110450.

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

## Acknowledgements

We thank C. Kadoch for the ARID1A CRISPR plasmid, D. Huntsman for the primary OCCC cultures, J. Conejo-Garcia for the genetic OCCC model, and S. Hua for technical assistance. This work was supported by US National Institutes of Health grants (R01CA160331, R01CA163377, and R01CA202919 to R.Z., R00CA194318 to B.G.B., R01CA131582 to D.W.S., R50CA221838 to H.-Y.T., and R50CA211199 to A.V.K.) and

US Department of Defense (OC140632P1 and OC150446 to R.Z.). Support of Core Facilities was provided by Cancer Centre Support Grant (CCSG) CA010815 to The Wistar Institute.

## Author contributions

S.W., N.F., T.F., B.G.B., P.H.P., H.-Y.T., and M.T. performed the experiments and analyzed the data. S.W., N.F., and R.Z. designed the experiments. A.V.K., H.-Y.T., and M.T. performed the bioinformatic analysis. L.Z., A.G., D.W.S., and R.Z. supervised the studies. S.W. and R.Z. wrote the manuscript. R.Z. conceived the study.

## Additional information

**Competing interests:** The authors declare no competing interests.

