## [Peer Review File · Nature Communications]

Reviewers' comments:

Reviewer #1 (Remarks to the Author):

In the manuscript by Wu et al, the authors investigated the mechanisms of acquired resistance to EZH2 inhibitors in ARID1A-mutant ovarian cancer cell lines. Prior work by the authors led to the discovery of EZH2 as a target in this disease, and this study leads to the discovery that an exchange of SMARCA4 for SMARCA2 is associated with acquired resistance to EZH2i. The functionality of the event is demonstrated through both gain- and loss-of function genetic experiments, which suggests that it is loss of SMARCA4 that drives this resistant state. Subsequent mechanistic studies highlight a role for apoptotic gene regulation as underlying this resistant state, and evidence is provided that combining BCL2 inhibition with EZH2 inhibition can overcome resistance in this disease.

Overall, I found the experiments to be rigorously performed, and leading to convincing demonstration of a mechanism of resistance. I also found the major finding of this study to be of high novelty, as it is distinct from other reported mechanisms of resistance to these agents. I support publication of this manuscript as is, without any additional experimental evidence being necessary. I only request that the authors provide additional discussion and/or speculation of the transcriptional mechanism underlying the loss of SMARCA4 and its exchange with SMARCA2.

Reviewer #2 (Remarks to the Author):

Comments to the Authors:

1. The authors have identified a potentially important mechanism of resistance to an EZH2 inhibitor.
2. While data have been replicated in two clones of this single cell line, the paper would be substantially strengthened if the authors could demonstrate that treatment of other ovarian clear cell carcinoma cell lines with GSK 126 produced a similar pattern of resistance.
3. As different parental clear cell ovarian cancer cell lines are likely to vary in their sensitivity to GSK126 possibly reflecting innate resistance to EZH2 inhibition, it would be reassuring to observe synergy between GSK 126 and BCL-2 inhibitors in multiple cell lines.
4. Ideally, clinical material and PDX models could be tested for a correlation between SMARCA4 levels and resistance to GSK 126 and susceptibility to a combination of GSK 126 and BCL-2 inhibitors.
5. In Extended Figure 2, SMARCA4 knockdown was performed in SKOV3 ovarian cancer cell line which was derived from a high grade serous ovarian cancer that is likely to have overexpression of EZH4 rather than mutation of ARID1A. Has ARID1A actually been sequenced in this cell line?

Point-by-Point Response to The Reviewers Comments

We sincerely thank both the Reviewers and the Editors for the constructive and thoughtful review provided for our manuscript. We are grateful for their shared appreciation of our manuscript as “*Overall, I found the experiments to be rigorously performed, and leading to convincing demonstration of a mechanism of resistance. I also found the major finding of this study to be of high novelty, as it is distinct from other reported mechanisms of resistance to these agents. (Reviewer 1)*”, and “*The authors have identified a potentially important mechanism of resistance to an EZH2 inhibitor (Reviewer 2)*”. All the comments raised are truly valuable to improve the manuscript. Correspondingly, we have strived to fully address their comments. I hope that there is no doubt that we have taken the Reviewers’ and Editors’ comments very seriously. We believe that by addressing the reviewers’ concerns we have produced a more solid and cohesive manuscript. A point-by-point response to the reviewers’ comments is detailed below with original comments italicized. Changes that directly address the reviewers’ concerns were denoted with vertical lines in the right margin in the revised manuscript. We hope the Reviewers and the Editors will find this manuscript to be much improved and suitable for publication.

Reviewers' comments:

Reviewer #1 (Remarks to the Author):

In the manuscript by Wu et al, the authors investigated the mechanisms of acquired resistance to EZH2 inhibitors in ARID1A-mutant ovarian cancer cell lines. Prior work by the authors led to the discovery of EZH2 as a target in this disease, and this study leads to the discovery that an exchange of SMARCA4 for SMARCA2 is associated with acquired resistance to EZH2i. The

functionality of the event is demonstrated through both gain- and loss-of function genetic experiments, which suggests that it is loss of SMARCA4 that drives this resistant state. Subsequent mechanistic studies highlight a role for apoptotic gene regulation as underlying this resistant state, and evidence is provided that combining BCL2 inhibition with EZH2 inhibition can overcome resistance in this disease.

Overall, I found the experiments to be rigorously performed, and leading to convincing demonstration of a mechanism of resistance. I also found the major finding of this study to be of high novelty, as it is distinct from other reported mechanisms of resistance to these agents. I support publication of this manuscript as is, without any additional experimental evidence being necessary. I only request that the authors provide additional discussion and/or speculation of the transcriptional mechanism underlying the loss of SMARCA4 and its exchange with SMARCA2.

Response: We are very appreciative for the positive comments by the reviewer. As suggested, we have now added discussion of the transcriptional mechanism underlying the loss of SMARCA4 and its exchange with SMARCA2 on page 11, paragraph 1.

Reviewer #2 (Remarks to the Author):

Comments to the Authors:

1. The authors have identified a potentially important mechanism of resistance to an EZH2 inhibitor.

Response: We are very appreciative for the positive comments by the reviewer.

2. *While data have been replicated in two clones of this single cell line, the paper would be substantially strengthened if the authors could demonstrate that treatment of other ovarian clear cell carcinoma cell lines with GSK 126 produced a similar pattern of resistance.*

Response: We thank the reviewer for the comment. We performed the requested experiments using an additional cell line. The results show that treatment of the *ARID1A*-mutated ovarian cancer cell line SKOV3 with GSK126 produced a similar pattern of resistance (**Supplementary Figure 3f**). Indeed, we observed a similar downregulation of SMARCA4 and upregulation of SMARCA2, which correlates with an upregulation of BCL2 in the SKOV3 GSK126 resistant cells (**Supplementary Figure 3g**).

3. *As different parental clear cell ovarian cancer cell lines are likely to vary in their sensitivity to GSK126 possibly reflecting innate resistance to EZH2 inhibition, it would be reassuring to observe synergy between GSK 126 and BCL-2 inhibitors in multiple cell lines.*

Response: We agree with the reviewer. We have now included the synergy analysis between GSK126 and BCL2 inhibitors in additional *ARID1A* deficient cell lines (**Supplementary Figure 4h and 4j**) and an *ARID1A* deficient primary clear cell carcinoma cultures (**Supplementary Figure 4l**).

4. *Ideally, clinical material and PDX models could be tested for a correlation between SMACA4 levels and resistance to GSK 126 and susceptibility to a combination of GSK 126 and BCL-2 inhibitors.*

Response: We thank the reviewer for the suggestions. Although we have tried very hard to generate PDX models, we have been unsuccessful in these efforts. However, we show that

GSK126 and BCL2 inhibitor are synergistic in suppressing the growth of ARID1A-deficient but not ARID1A-proficient primary ovarian clear cell carcinoma cultures (**Supplementary Figure 4k-m**).

5. In Extended Figure 2, SMARCA4 knockdown was performed in SKOV3 ovarian cancer cell line which was derived from a high grade serous ovarian cancer that is likely to have overexpression of EZH2 rather than mutation of ARID1A. Has ARID1A actually been sequenced in this cell line?

Response: We thank the reviewer for the comments. Although SKOV3 was originally described to be a cell line derived from a high-grade serous ovarian cancer, recent genomics studies suggest that the cell line does carry an *ARID1A* mutation^{1,2}. Consistently, clear cell-like histology has been observed when SKOV3 is grown as a xenograft³. Thus, it has been suggested that the SKOV3 cell line is unlikely a high-grade serous cell line, but instead a cell line of clear cell origin^{1,2}.

Cited references:

1. Anglesio MS, *et al.* Type-specific cell line models for type-specific ovarian cancer research. *PLoS One* **8**, e72162 (2013).
2. Domcke S, Sinha R, Levine DA, Sander C, Schultz N. Evaluating cell lines as tumour models by comparison of genomic profiles. *Nat Commun* **4**, 2126 (2013).

3. Shaw TJ, Senterman MK, Dawson K, Crane CA, Vanderhyden BC. Characterization of intraperitoneal, orthotopic, and metastatic xenograft models of human ovarian cancer. *Mol Ther* **10**, 1032-1042 (2004).

REVIEWERS' COMMENTS:

Reviewer #2 (Remarks to the Author):

1. The paper has been strengthened by finding a similar mechanism of resistance in a second cell line and by demonstrating synergistic interactions between GSK126 and BCL2 inhibitors in more than one cell line.

2. While establishing PDX models in a relatively rare tumor type is a challenge, SKOv3 grows well as a xenograft and TOV21G cells may as well. Other ARID1A mutant cell line are available. As this is a paper regarding a combination of targeted therapies and a mechanism of resistance, in vivo data in two xenografts lines would be expected by most, if not all, respected journals. If the combination of agents is not effective against xenografts, the mechanism of resistance, while novel, may not be important.

3. It would still be desirable to sequence ARID1A in the SKOv3 that the investigators are using. In our own laboratory's experience SKOv3 does not exhibit histological clear cell characteristics in xenograft models.

Point-by-Point Response to The Reviewers Comments

Reviewer #2 (Remarks to the Author):

1. *The paper has been strengthened by finding a similar mechanism of resistance in a second cell line and by demonstrating synergistic interactions between GSK126 and BCL2 inhibitors in more than one cell line.*

Response: We thank the reviewer for the comments.

2. *While establishing PDX models in a relatively rare tumor type is a challenge, SKOV3 grows well as a xenograft and TOV21G cells may as well. Other ARID1A mutant cell line are available. As this is a paper regarding a combination of targeted therapies and a mechanism of resistance, in vivo data in two xenografts lines would be expected by most, if not all, respected journals. If the combination of agents is not effective against xenografts, the mechanism of resistance, while novel, may not be important.*

Response: We regret that the reviewer missed the *in vivo* combination data in xenograft model in Supplementary Figure 5D. Indeed, we showed that the combination of agents is effective against ARID1A-inactivated xenografts.

3. *It would still be desirable to sequence ARID1A in the SKOV3 that the investigators are using. In our own laboratory's experience SKOV3 does not exhibit histological clear cell characteristics in xenograft models.*

Response: We thank the reviewer for the comments. Indeed, the SKOV3 cell line used in the current study in the lab has been characterized for ARID1A status in our previous publication (*Bitler et al. Nature Medicine, 21: 231-8*). For example, only a truncated form of ARID1A is expressed in SKOV3 as showed in Figure 2A of our recent publication (*Bitler et al. Nature Medicine, 21: 231-8*).